# Health Education Initiatives for People Who Have Experienced Prison: A Narrative Review

**DOI:** 10.3390/healthcare12020274

**Published:** 2024-01-22

**Authors:** Patrícia de Paula Queiroz Bonato, Carla Aparecida Arena Ventura, Réka Maulide Cane, Isabel Craveiro

**Affiliations:** 1Unidade de Ensino e Investigação de Saúde Pública Global, Global Health and Tropical Medicine (GHTM), Associate Laboratory in Translation and Innovation Towards Global Health (LA-REAL), Instituto de Higiene e Medicina Tropical (IHMT), Universidade NOVA de Lisboa (UNL), Rua da Junqueira 100, 1349-008 Lisboa, Portugal; patricia.bonato@usp.br (P.d.P.Q.B.); rica.cane@gmail.com (R.M.C.); 2Ribeirão Preto College of Nursing, University of São Paulo (USP), Rua Prof. Hélio Lourenço 3900, Ribeirão Preto 14040-902, Brazil; caaventu@eerp.usp.br; 3Instituto Nacional de Saúde (INS), Ministério da Saúde (MISAU), Estrada Nacional EN1, Bairro da Vila—Parcela no 3943, Distrito de Marracuene, Marracuene 264, Província de Maputo, Mozambique

**Keywords:** health education, educational initiatives, prisons, prisoners, people released from prison

## Abstract

Due to the selectiveness of criminal systems and the context of social vulnerability, there is a high prevalence of health problems among individuals with a history of incarceration. When there is an insufficient level of health care, prior clinical conditions can worsen, and health education can be a response to this problem. Health education is a process of building health knowledge that is intended to facilitate thematic appropriation by the population that enables people to access, understand, and use health-related information for health improvement. In the context of criminal justice, health education can contribute to the successful transition of people who have experienced prison from their custody to the community setting. This study aimed to identify, synthesize, and critically evaluate peer-reviewed evidence concerning health education initiatives developed during or after incarceration aimed at people released from prison. A narrative review methodology was used to analyze 19 studies about health education interventions for prisoners or people who were arrested. Initiatives were identified in five countries, which showed differences in approaches, with motivational interviewing and group sessions standing out in the studies. All of them were grouped into the following themes: HIV and other sexually transmitted infections, alcohol, opioids and other substances, tuberculosis, and women’s health. We have not performed a quality assessment of the studies included (using checklists such as PRISMA, AMSTAR, or SANRA) as this study is a narrative review and was not intended to be a systematic review or meta-analysis. This review has the potential impact of informing future health education initiatives and policies for individuals transitioning from prison.

## 1. Introduction

By improving people’s access to health information and their ability to use it in personally meaningful ways, health literacy is crucial to autonomy and health promotion. Studies reveal that limited health literacy skills in people in conflict with the law serve to intensify experiences of disenfranchisement, isolation, and shame [1,2]; in this context, health literacy is recognized as a type of intervention for health promotion [3]. The aspect of literacy gains relevance and priority in the context of criminal justice, as autonomy in making informed decisions is fundamental for what international literature calls a “successful transition from custody to the community”, which demands engagement in access to health services [4,5].

Health literacy refers to the ability of the public to access, understand, communicate, and use health-related information in a way that promotes and maintains health [6]. The concept of literacy or health literacy has gained prominence in public health as it enables innovation in individual care. The World Health Organization (WHO) defines literacy as “a set of cognitive and social skills, as well as an understanding and use of information that promotes and maintains good health” [7].

The effects of incarceration on individual health are well known, but the release also has long-term effects and constitutes a high concentration of risks. People who have been arrested are 12 times more likely than the general population to die within the first two weeks of release from any cause [8,9], as well as having disproportionately higher rates of hepatitis C [10].

In this context, overdose is the most common cause of death after release from prison [11], and psychostimulants are the most common substance involved in such deaths [12]. 

In addition to pre-existing social vulnerability to incarceration, this population is also affected by several social determinants of health, including unemployment, histories of intra-family and community violence, abuse of legal and illicit drugs [13,14], and low educational attainment. As most incarcerated individuals often return to their communities, healthcare delivery during incarceration plays a substantial role in the health of communities at all levels [8].

Sustainable Development Goal (SDG) 3 aims to ensure healthy lives and promote well-being for all at all ages, particularly for vulnerable populations, including people who have experienced prison. As such, it is imperative to develop coordinated health programs across correctional and community programs [9,15,16] to support the successful re-entry of these people who have experienced prison into their communities. Thus, health education can be a viable approach at the point of transition to the community or even after liberation. 

It is, therefore, essential that health education is provided to help people recognize the importance of health care in other spheres of life that are involved in the transition to a community. As a result of knowledge of how to make informed decisions, as well as access to health services, people who return to social life will experience greater well-being, which will positively affect their employment, family, and personal relationships in the future.

Therefore, this narrative literature review aims to identify, synthesize, and critically evaluate the scientific evidence on health education initiatives aimed at adults who are about to or recently released from prison. 

## 2. Materials and Methods

This narrative review was guided by the following question: “What are the existing health education initiatives developed for adults (adults over the age of 18) recently released from prison or about to be released?”. Typically, narrative literature reviews present a more open theme without a rigid methodology and are used with an exploratory objective to identify studies produced on a particular topic without rigorous assessment of the methodology and quality of the studies.

The following question has been formulated using the PICo strategy: (P: patient or problem; I: intervention; Co: context). In the context of this narrative review, a patient (P) is a person who has experienced prison and who has already been released or who is in the process of transitioning back into society; (I) refers to the health education intervention, which in the context of the present study varied from focus group discussions and motivational interviews to traditional educational programs (lectures, readings, dramatizations, and group discussions with didactic content); and context (Co) refers to life prison experience [17]. 

The search was carried out in seven (07) electronic databases: PubMed, Embase, CINAHL, Socindex, Scopus, Web of Science, and the Virtual Health Library (VHL/LILACS), using the descriptors: (“Post-release” OR “post-release” OR “Post-Incarceration” OR “Post Incarceration” OR “after incarceration*” OR “Post-Carceral” OR “Post Carceral” OR “after release” OR “Ex-prisoner*”) and (“Health Education” OR Education*), as well as other derivations associated with search terms according to the search criteria for each database. Publications in English, Portuguese, and Spanish were included.

The inclusion criteria for the selected articles were primary studies using qualitative and/or quantitative mixed or theoretical methods, retrieved without limitation on the year of publication. On the other hand, articles without open access or literature review, course completion works, editorials, study protocols, or descriptions of work not yet implemented were adopted as exclusion criteria. Two independent reviewers (PQB and CAV) performed the narrative review, and the researchers’ disagreements over whether an article should be included in the review were resolved by mutual agreement.

## 3. Results

### 3.1. Search Results

In total, 1462 studies were initially identified. First, duplicate works were excluded (*n* = 484). The titles of the remaining 978 articles were reviewed using the inclusion criteria. In this second stage, 950 articles were excluded, 934 of these because they dealt with other topics, 10 because they referred to review studies, and 6 because they addressed interventions in the context of juvenile justice.

Thus, as a final step, 35 studies were read in full, but 16 of them did not meet the inclusion criteria and were excluded for the reasons set out below in the flowchart of the search strategy (Figure 1). The process of inclusion and exclusion of articles took place between February and August 2023.

### 3.2. Studies Characterization

The following data refer to the general characteristics of selected studies. This review included 19 studies in total. We classified the studies according to authors, year of publication, country, theme and moment of intervention, sample (participants), objectives, methods, and main findings (Table 1).

The oldest study publication included in this review dates back to 1997 (5.2%). Five studies were published between 2001 and 2008 (26.3%), seven studies were published between 2014 and 2018 (42.1%), and six between 2019 and 2022 (31.5%). Regarding the country of origin, the majority (13 studies) were developed in the United States (68.4%), two in Australia (10.5%), two in Cuba (10.5%), and the rest are equally distributed (5.2% in each) from Brazil and South Africa. 

Of the total number of studies selected (*n* = 19), the majority were quantitative studies (12), others were qualitative (2), and the rest were classified with a mixed methodological approach, quantitative and qualitative (5). Regarding the health interventions described in the studies, most (11) were developed in the context of the transition from prison to the community [18,19,20,21,22,23,24,25,26,27,28], two (02) were initiated after people returned to social life [29,30], and six (06) began during incarceration and were continued after arrest [31,32,33,34,35,36].

Finally, it is relevant to highlight that five works (5) were focused on women’s health, approaching topics such as reproductive health, cervical cancer, and HIV risk reduction. A more detailed synthesis of the selected studies’ content can be found in Table 1.

**Table 1 healthcare-12-00274-t001:** Characterization of selected studies.

Authors, Year, and Place of Study	Theme and Moment of Intervention	Study Objectives	Methods	Main Results
Hunt et al. (2022, EUA) [18]	Women’s health; moment of intervention: transition to the community	To develop a sustainable pre-release education program to reduce the risk of opioid overdose post-release in female inmates in a rural county jail.	Qualitative study The authors developed a re-entry program entitled Healthy Outcomes Post Release Education (HOPE), using the Roy adaptation model as a theoretical framework.	The implementation of a successful re-entry program for the vulnerable female incarcerated population has the potential to reduce the risk of opioid overdose death and negative health outcomes post-release and provide an opportunity for adapting culture and starting other supportive re-entry type programs.
Geana et al. (2021, EUA) [29]	Women’s health; moment of intervention: after release	Enhance health literacy through the development and testing of an intervention (mHealth).	Mixed study. A website was developed for the intervention based on four themes identified in previous research (cervical cancer, breast health, reproductive health, and STIs).	This intervention may increase the health literacy of women participating in the study and may have positive health and behavioral effects.
Winkelman et al. (2021, EUA) [31]	Alcohol, opioids, and other substances; moment of intervention: transition to the community and after release	To examine the acceptability, feasibility, and preliminary clinical outcomes of a smoking cessation intervention for individuals who are incarcerated.	Randomized clinical trial. Two groups: (a) Nicotine replacement therapy (NRT): received 1 h of smoking cessation counseling in jail, a supply of nicotine lozenges upon release, and up to 4 telephone counseling sessions after release; (b) brief health education (BHE): received 30 min of general health education in jail.	Initiation of counseling plus NRT during incarceration and continuing after release is feasible and acceptable to participants and may be associated with reduced cigarette use after release. Participants expressed a positive impression of the in-jail counseling or education.
Banta-Green et al. (2020, EUA) [32]	Alcohol, opioids, and other substances; moment of intervention: transition to the community and after release	To evaluate an opioid use disorder (OUD) treatment decision-making intervention to determine whether the intervention increased medications for opioid use (MOUD) disorder initiation post-release.	Observational retrospective cohort study. The intervention: (a) included education on OUD/MOUD; (b) explored people’s perceptions and history of use of MOUD; (c) provided a motivational-interviewing-informed approach to evaluating the pros and cons of each medication, given a person’s preferences and life circumstances; and (d) helped identify specific next steps towards initiating MOUD, if selected as the desired treatment.	Those who received the intervention were significantly more likely to start medication during the first month after release from prison but not in the following months. The short-term nature of the effect observed in this study supports efforts to determine how to support people in staying on medications for opioid use disorder as long as they wish to continue.
Brousseau et al. (2020, EUA) [19]	Women’s health; moment of intervention: transition to the community	To assess the efficacy of motivational interviewing as an individualized intervention to increase the initiation of contraceptive methods while incarcerated and continuation after release in female inmates.	Randomized control trial (RCT). Treatment group: women assigned to motivational interviewing. Control group: women assigned educational videos with counseling.	There was higher initiation of contraception in the motivational interviewing group compared to educational video group. After controlling for significant treatment group differences, motivational interviewing did not increase contraceptive initiation, nor did it decrease pregnancies.
Wiersema et al. (2019, EUA) [33]	HIV and other sexually transmitted infections Moment of intervention: transition to the community and after release	Describe the adaptation of an evidence-based intervention to influence sexual behavior.	Quantitative study. The risk reduction intervention delivered through two 1.5 h sessions. Participants completed a baseline survey before the first group session and a postintervention survey a few days after the last group session. The baseline survey contained questions to assess HIV knowledge, HIV risk, and health-promoting behaviors.	Participants showed significantly improved knowledge of HIV. 90 days after release from prison, participants reported improved “CLEAR thinking”, reduced risk behaviors, and improved health-promoting behaviors.
Corsino et al. (2018, Brazil) [20]	Women’s health; moment of intervention: transition to the community	To analyze the impact of educational action on HPV carried out with women in prison in Mato Grosso.	Experimental, inferential comparative study of the “before and after” type.	Carrying out the educational action contributed significantly to the information the women had about HPV. Educational actions are efficient forms of information that equip women in decision making. There is a clear need for educational interventions in prisons to promote health and prevent disease, which can reduce the vulnerability of individuals.
Staton et al. (2018, EUA) [34]	Women’s health; moment of intervention: transition to the community and after release	Examine the delivery of an HIV risk reduction intervention focused on prevention education (NIDA Standard) and an enhanced individualized risk reduction intervention (MI-HIV) in rural prisons to reach high-risk rural women who use drugs. Specifically, the study examines short-term outcomes 3 months after release from prison for high-risk rural women.	Quantitative study. Consultative sampling, screening, and face-to-face interview approach to recruiting hard-to-reach, out-of-treatment rural drug-using women in three Appalachian-area prisons. Analyses included descriptive statistics, multinomial logistic regression, and stepwise regression to identify significant correlates of recent and past injecting drug use compared to never injecting.	HIV education interventions can be associated with risk-reduction behaviors. Decreases in HIV risk behaviors were observed at follow-up across all conditions. Although participants in the MI-HIV group experienced reductions in outcomes compared to the NIDA Standard group, these estimates did not reach significance. Educational interventions about HIV can be associated with risk reduction behaviors.
Thornton et al. (2018, EUA) [21]	HIV and other sexually transmitted infections; moment of intervention: transition to the community	Evaluate the ECHO New Mexico Peer Education Project, focusing on the short-term impact of training.	Mixed study. Intensive 40 h training (delivered over five consecutive days) for incarcerated people. Ten-hour workshop, conducted for 2 to 3.5 h over 3 to 5 days. Peer educators conduct the training in its entirety, and students receive short educational presentations and are engaged in a variety of learning activities. Three-hour educational session at the prison’s central reception for all men entering the prison system.	Significant changes were observed in knowledge, attitudes, behavioral intention, and self-efficacy. Programs like this have the potential to help slow a deadly HIV epidemic by educating individuals to reduce the risk of acquisition and transmission, both in prison and upon return to their communities.
Williams et al. (2018, EUA) [30]	HIV and other sexually transmitted infections; moment of intervention: after release	To determine the efficacy of the cognitive-behavioral intervention in improving the sexual health of minority men after jail release.	Randomized controlled trial. Participants were assessed and tested for three sexually transmitted diseases (STDs) and HIV at baseline and 3 months post-intervention and followed up for 3 more months.	The intervention group’s STD risk knowledge, partner communication about condoms, and condom application skills improved. A tailored risk reduction intervention for men with infection histories can affect sexual risk behaviors.
Hernandez et al. (2016, Cuba) [22]	HIV and other sexually transmitted infections; moment of intervention: transition to the community	To perform an educational intervention study in Mar Verde Penitentiary Center with the purpose of diminishing syphilis incidence.	Qualitative study. The educational work was based on a methodological manual for training prosecutors who work with their peers in prisons modified by the authors for the prevention of STIs/HIV/AIDS. The promoters carried out a set of prevention activities, such as video debates, face-to-face conversations and counseling, knowledge meetings, conferences, and get-togethers.	With the educational intervention, it was possible to reverse the epidemiological situation of syphilis, since in 7 days, 42 promoters were trained who worked with their peers and who, with support, were able to develop an educational program that, in less than 2 years, reduced the incidence of infection by 70.6%. It is recommended to promote the implementation of educational interventions in other penal institutions for their usefulness in the improvement of the health of its inmates and as an important contribution to the system of medical care in these centers.
Fogel et al. (2015, EUA) [35]	Women’s health; moment of intervention: transition to the community and after release	To test the efficacy of an adapted evidence-based HIV–sexually transmitted infection (STI) behavioral intervention (Providing Opportunities for Women’s Empowerment, Risk-Reduction, and Relationships, or POWER) among incarcerated women.	Randomized trial intervention participants attended 8 POWER sessions; control participants received a single standard-of-care STI prevention session. Participants were followed up at 3 and 6 months after release. The intervention efficacy was examined with mixed-effects models.	Women were followed up 3 and 6 months after release. There were decreases in HIV risk behaviors and intention to begin treatment, but the differences between the groups were not statistically significant. A multi-session HIV-STI prevention intervention adapted for and delivered to women in prison can significantly reduce sexual risk behaviors and increase protective behaviors after re-entry into the community. Prisons provide an opportunity to deliver behavioral interventions to a population at high risk of acquiring or transmitting HIV and STIs.
Sifunda et al. (2008, South Africa) [23]	HIV and sexually transmitted infections; moment of intervention: transition to the community	To test the effectiveness of a systematically developed health education intervention targeting soon-to-be-released prison inmates on psychosocial determinants of reducing risky sexual behaviors.	Nested experimental design. Within each of four selected prisons, there was both a control and an experimental group. The experimental groups were divided between those who were instructed by an HIV-positive peer educator and those who were instructed by an HIV-negative peer educator. The study used a pre-test, a post-test prior to release from prison, and a 3- to 6-month community follow-up test as evaluation measurements for all participating inmates.	This study was one of the few that explored the use of theoretical approaches in designing health education interventions aimed at inmates and probably the only one conducted in sub-Saharan prison conditions. Curriculum content should be simple and targeted at the desired areas of change to ensure that even people with low literacy levels are able to understand all program concepts. The results demonstrate an already established benefit of small group skill-building risk-reduction programs. The lessons learned and the issues that emerged during the adaptation of the intervention underscore the need for linguistically and socioculturally appropriate, locally designed programs that can shed some light and lead to greater understanding of these contextual differences.
Valle Yanes et al. (2008, Cuba) [24]	HIV and other sexually transmitted infections; moment of intervention: transition to the community	Evaluate the effectiveness of an educational intervention to raise awareness of HIV.	Quantitative study. A pre-experimental study (before–after) was carried out. Five 45 min meetings were held with each group of 25 prisoners to implement the educational program. Six weeks later, the initial instrument was applied again to verify knowledge acquisition.	Before the educational intervention, 71% of the inmates surveyed had an “unacceptable” level of knowledge, as they did not recognize the three most common ways to prevent infection. After, that number dropped to 53%. After the educational intervention, participants acquired acceptable and moderately acceptable levels of knowledge about HIV in terms of the most common ways to become infected, prevention, and vulnerability to infection.
Minc et al. (2007, Austrália) [25]	HIV and other sexually transmitted infections; moment of intervention: transition to the community	To deliver engaging, relevant, and clear health messages to prison inmates, ex-inmates, and families in relation to HIV, hepatitis, and sexual health.	Interventional radio program/qualitative study. A community restorative center that broadcasts a weekly half-hour radio program to prisoners and the community to provide support to prisoners, exprisoners, and their families.	The most popular programs were those that either featured a particular goal or an inmate’s, ex-inmate’s, or family’s lived experience of the prison system rather than programs specifically devoted to health. The vast majority of prisoners and stakeholders consider a radio program for prisoners to be both a relevant and useful strategy to complement existing written health promotion material and address poor literacy levels.
White et al. (2003, EUA) [26]	Tuberculosis; moment of intervention: transition to the community	To describe the educational process provided to inmates in jail, including developing the educational protocol, hiring, and training of staff members. To describe the application of an educational intervention in a jail setting.	Qualitative study. Development of the protocol and educational process provided to prisoners. The individual educational session lasted 10 to 15 min and was led by community health workers, who presented themselves as representatives of an external agency (the University). The materials were delivered at the end.	The project was very successful in educating inmates and motivating prisoners to continue care after release. As compared with 3% clinic visit rates before the project began, 24% to 37% of inmates receiving education through the tuberculosis prevention project completed tuberculosis clinic visits depending on intervention arm. Jails provide a rewarding opportunity to work with persons at high risk for a number of diseases. The jail setting is a safe and relatively quiet place and provides an opportunity to educate a population in great need of intervention.
Grinstead et al. (2001, EUA) [27]	HIV and other sexually transmitted infections; moment of intervention: transition to the community and after release	Reduce risky sexual and drug-related behavior and encourage the use of community resources following release.	Quantitative study. Eight sessions over two consecutive weeks, totaling 16 to 20 h. During the seventh and eighth sessions, service providers from participating release counties provided information and made appointments for post-release services.	HIV-positive inmates were well informed about the facts of transmission and prevention. Need for transitional case management to facilitate use of community resources, especially drug and alcohol treatment.
Crundall and Deacon (1997, Australia) [36]	Alcohol, opioids, and other substances; moment of intervention: transition to the community and after release	Determine the effect of an alcohol education course on alcohol consumption, drinking group, disruptive behavior, criminal activity, family relationships, how they use their time, general health, ability to cope and take responsibility.	Mixed study (quanti-qualitative). Treatment group: those who completed the course. Control group: others who had not done the course.	The prisoners attending the course showed improvements in all dimensions when compared to the control subjects. Involvement in the course influenced more than just the drinking and offending behaviors of the prisoners, with indirect gains in terms of health, personal dispositions, and relationships.
El-bassel et al. (1997, EUA) [28]	Women’s health; moment of intervention: transition to the community	Enhance sexual safety and reduce risky behavior.	Mixed study. Pre-intervention: focus groups with 20 women (arrested and released). Intervention: eight weekly 1.5 h sessions included simulations, minimal didactic content, and reading tasks between sessions. During the first two months after release, there were eight “boosting” sessions, the first being four days after release.	By learning about HIV rates among women prisoners, participants gained a better understanding of how HIV affects children, families, and communities. As a result of this knowledge, followed by training to achieve social support, self-efficacy can be enhanced.

## 4. Discussion

### 4.1. Transition from Prison to the Community and Health Education Interventions

Most of the studies described in this review were developed in the context of the transition from prison to the community. As stated by previous authors [37], prisoner re-entry includes many processes that begin before the individual is released from prison, as well as experiences at the moment of release and during the first few years in the community. Educational interventions allow the provision of new cognitive, social, and health education skills. Several authors have reported that good health in prison is associated with a reduced likelihood of recidivism [38]. Thus, educational interventions and research involving prison health care generally recognize that incarceration is an opportunity (sometimes the first) for medical care [26,28], diagnosis, and treatment of illness.

Despite that, internationally, numerous challenges were reported in the criminal justice setting and the intersection with health care during and after prison, highlighting limited financial resources, the absence or underdevelopment of reintegration programs, and those focused on adaptation, especially for women [18], and the lack of knowledge or lack of control over the release date or even short lengths of stay [39]—factors that result in interventions that are sometimes briefer and more flexible [31].

### 4.2. Women’s Health and Techniques for Health Education and Promotion

Nearly one-third of the studies included in this review focused on health education for women. Previous literature showed that women with criminal records are more likely to suffer from chronic diseases such as cancer, hypertension, heart disease, and diabetes [40], thus requiring more attention from academics and governments in the scenario of public health focused on prison health. Even though research on the general health of criminalized women is needed, it is limited and almost nonexistent [41,42].

Some authors argue that ideas relating to health promotion in prisons, especially in women’s prisons, are in development. Such include a more participatory approach, using community development methods that can contribute to making women prisoners more health literate and more confident to look after their health and the health of their children (particularly babies born in prison) [42]. 

On the other hand, few studies indicate that women leaving prison prefer multimedia interactions [43]. In contrast, other studies suggest that health information delivered via video is more likely to lead to a change in preventive behavior than traditional methods (pamphlets), especially for audiences with less health education [44].

### 4.3. Focus Group Discussions, Motivational Interviews, and Health Education Interventions

Focus group discussions and motivational interviews were used in some of the interventions reported by our review. In terms of the intervention technique, motivational interviews have shown to be suitable for incarcerated populations [45,46], as evidenced by research conducted on urban women involved in criminal justice, which indicated a reduction in risky sexual behaviors as well as drug abuse [46].

When it comes to substance abuse, in general, motivational interviewing is regarded as an effective tool for treating this health issue since more traditional treatment during and after incarceration may be limited by barriers to access, both inside and after prison [47]. Although studies using a motivational interview approach for alcohol consumption education for prisoners were not included in the present review, part of the literature indicates that such an approach is more effective than educational interventions [48]. It should be noted, however, that there is a low quality of literature on motivational interviewing on this topic due to the lack of measurements of motivation to change and duration of treatment, among other parameters [47].

### 4.4. Alcohol, Drug Abuse, and Health Education Interventions

Apart from alcohol consumption, some of the studies in this review dealt with issues linked to opioids and other substances. The association between substance use and crime is very common but complex [49]. Opioid use disorder is a very common problem in prisons, which can be explained by drug criminalization and, therefore, by people’s involvement and use of illegal drugs. An estimated one-third of criminal justice-involved individuals with substance use disorders receive treatment within 1 year of their release [50].

In this review, we observed that most of the interventions aimed at preventing substance abuse were led during imprisonment and that, in at least three of them [18,32,36], participants were followed up after release to determine the effects of the intervention. This is aligned with other authors who state that it is important to adapt the programs to the needs and individual characteristics of each offender in order to achieve a successful intervention [49].

### 4.5. Lack of Educational Interventions Linked to Noncommunicable Diseases

Even though people deprived of liberty have an increased risk of developing noncommunicable diseases by 1.2 to 1.8 times more often compared with the extramural community, and cardiovascular diseases are among the most common causes of death in prisons [51], especially hypertension and smoking [52,53], we observed a lack of educational interventions performed for this particular issue. 

Tuberculosis is also a particular health concern in prison settings, and it often occurs at a tenfold higher rate in prisoners than in the general population [28]. The impact of this on public health is often related to the interruption of treatment initiated in prison as a result of the release of prisoners. 

As stated by previous authors [54], prison health policies that combine prevention and treatment measures via physical activities, healthier diet, and smoke-free prisons can contribute to supporting the sustainable development goal 3.4 of “reducing one-third of the incidents of premature mortality from noncommunicable diseases”. Additionally, more research is needed to provide scientific-based evidence that can be used by stakeholders to improve ongoing prison health policies and strategies.

### 4.6. HIV and Other Sexually Transmitted Infections 

Several of the studies approached in this review included sexual health interventions initiated in prison and whose effects could be monitored or evaluated after release. The increase in the incidence of human immunodeficiency virus (HIV) infections in prisons has historically been another major health problem, which is why the prison environment is considered an important focus for prevention, screening, and referral to healthcare [55]. It is for no other reason that this topic was the most discussed in studies.

We observed that most of the research conducted on sexual health education focused on HIV infections as the primary concern, and literature [43] has indicated that providing general education and case management services enhances retention in treatment.

A major challenge identified is the continuity of care initiated in prison, as post-release infections are extremely difficult to measure outside of carefully constructed, funded, and controlled research settings [44]. Due to this, research supports the need to manage the transition from prison to society to facilitate the use of community resources, especially those associated with drug/alcohol treatment combined with sexual health care [46].

Almost all of the studies included in this review described the group intervention approach as the main strategy for the prevention of risky sexual behaviors, using health education sessions in the form of lectures and presentations. Prison peer education was a crucial intervention strategy used to deliver health education, contributing to reducing behavioral risks in participants [44]. 

Nevertheless, the majority of studies addressing peer education interventions in prisons still lack adequate methodological quality. Authors argue, however, that prevention strategies that rely solely upon education may not be sufficient to reduce sexual risk among these populations and that behavioral interventions may result in more concrete habit changes [56], especially when considering the biomedical needs of women incarcerated who are HIV/AIDS-positive [57]. 

### 4.7. Limitations

The results of this narrative review should be interpreted with caution due to specific study limitations of the literature included, such as the small sample size of some of the quantitative studies (*n* < 100).We did not perform a quality assessment of the studies included (using checklists such as PRISMA, AMSTAR, or SANRA) as this study is a narrative review and was not intended to be a systematic review or meta-analysis.The limitation of the study language was a limitation in the present study.

## 5. Conclusions

In this review, we identified relevant health education initiatives around the world that target individuals undergoing the transition from prison to community life or recently released from prison. The implementation of educational interventions for prisoners through lectures, discussions, reading tasks, and group activities were critical aspects for achieving successful results. 

As seen, educational interventions develop new cognitive, social, and educational skills that are essential for this public, not only due to the history of social vulnerability that generally identifies this population but also due to the fact that health care within a prison environment implies a reduced likelihood of repeating criminal behavior. When people received health education instructions following their release, they were more likely to initiate medication treatment in the first month following their release. Re-entry programs based on health education have the potential to reduce the risk of death due to opioid overdose in females. 

This narrative review unveils the need for more research in health education interventions focused on people who are in prison or have experienced prison, aiming particularly at public health topics where gaps prevail (such as noncommunicable diseases).

Considering the implications for practice, this present work has the potential impact of informing future health education initiatives and public policies for individuals who have experienced prison, particularly those in the pre-release moment. In the context of criminal justice, implementing health education initiatives such as those discussed in this study will facilitate the development of possibilities for social reintegration through health, including facilitating the work of both justice and health professionals.

## Figures and Tables

**Figure 1 healthcare-12-00274-f001:**
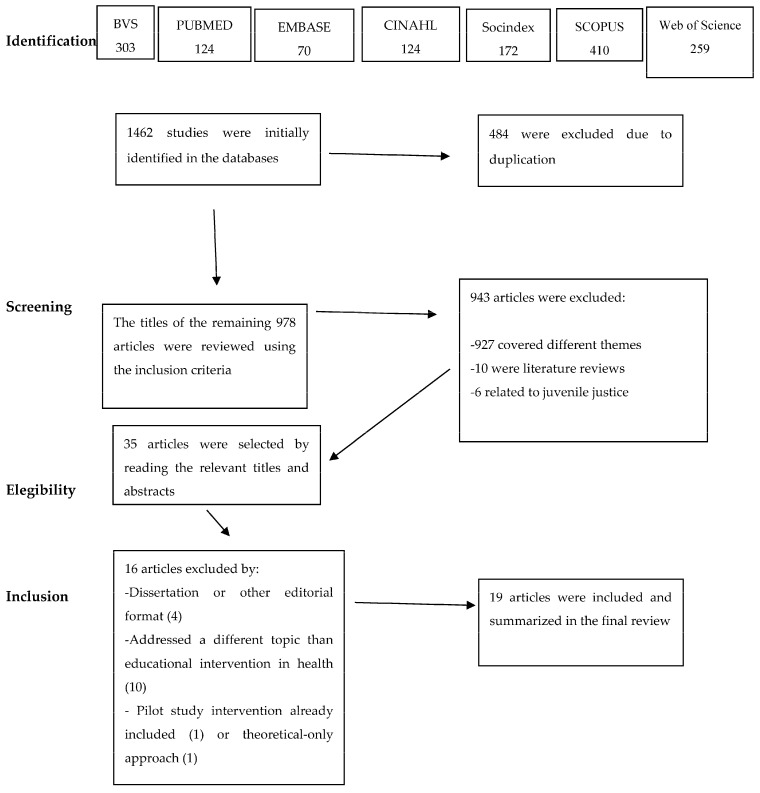
Flowchart of the search strategy and article selection phases.

## Data Availability

Not applicable.

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
