# Peer review of "Health Education Initiatives for People Who Have Experienced Prison: A Narrative Review"

_healthcare, 2024, doi:10.3390/healthcare12020274_

Round 1

Reviewer 1 Report (New Reviewer)

Comments and Suggestions for Authors

Dear editor,

Thank you for giving me the opportunity to review the manuscript. The topic of the manuscript was interesting to me. The article is well written. In the following, some comments are presented as suggestions for improving the manuscript.

Abstract

-The definition of health promotion is mentioned in the abstract, but health education interventions are discussed throughout the text. The definition of health education should be mentioned in the abstract.

Introduction

-Change the position of the first and second paragraphs in the introduction. It confuses the reader why the introduction started with health literacy?

-It is suggested to mention reliable statistics regarding the need of prisoners for health education and its effects.

Method

-How was the disagreement between researchers to enter or leave the study managed?

Results

-In Table 1, it is better to present the studies in the order of publication year (newest to oldest).

Discussion

-In the discussion section, the details of the included studies should also be stated, for example, in the women's health section, what activities have been done in the included studies related to women's health and should they be compared?

This can be seen more or less in other paragraphs of the discussion.

-The limitation of the study language should also be mentioned in the limitations of the study.

Author Response

Dear Reviewer 1,

We would like to express our sincere thanks for the time and effort you have dedicated to reviewing our article for publication in Healthcare Journal. Based on your comments, we have made the following changes to the manuscript: 

-The definition of health education was mentioned at the beginning of the summary and is highlighted (please see lines 12-13)

-The position of the first and second paragraphs in the introduction was changed, according to the editor's suggestions.

- We appreciate your suggestion regarding the need for health education and the effects of it on prisoners, but at the present time it is not possible to provide statistics regarding this topic, precisely because it is a topic that is not widely discussed. To fill this gap, we illustrate the harm of incarceration to people's health in the Introduction in order to draw attention to the need to take care of these demands, and also emphasize the lack of health literacy that has been highlighted by prior studies and which reinforces this need.

-The researchers' disagreements over whether an article should be included in the review were resolved by mutual agreement (please see lines 91-92)

-In Table 1, the studies were presented in the order of publication year, newest to oldest.

-The limitation of the study language was included as a limitation (please see line 279)

Reviewer 2 Report (New Reviewer)

Comments and Suggestions for Authors

This is an interesting piece of genuine research that focuses on a rather neglected aspect of public health domain, i.e. health education and an essentially vulnerable population, i.e. released prisoners, that deserve our attention.

However there is a number of issues that need to be addressed before this manuscript is considered for publication.

First, it appears that there are several language problems and the text needs to be proofread by a native speaker. Beyond the language issues, it seems that the manuscript was submitted in a hastiness with punctuation errors (e.g. in Line 23 you are missing the full stop after the meta analysis), didn´t perform (L. 21) have not performd, the gaps (L.23)the gap.

Second, the abstract starts with a profoundly repeated definition of health provided by the WHO (L 11-12). This is double problematic: a) there is reference to this quoted definition, b) it is a trivial way to start with. Consider adding a sentence or two at the beginning to provide broader context on the challenges faced by individuals transitioning from prison to the community and the importance of health education in this context. The objectives are well-stated, but you may want to explicitly mention the significance of the study. Why is it important to identify, synthesize, and critically evaluate these health education initiatives? While it's mentioned that a narrative review methodology was used, you might briefly explain why this methodology was chosen and its advantages for this particular study. This could help readers understand the approach better. You've touched on the countries where initiatives were identified and mentioned motivational interviewing and group sessions. If there are key themes or patterns that emerged from the studies, you could briefly highlight those in the abstract. The decision not to perform a quality assessment is explained, but you might want to briefly mention that this is a common approach in narrative reviews, as opposed to systematic reviews or meta-analyses. Consider adding a sentence about the potential impact of the review on informing future health education initiatives and policies for individuals transitioning from prison. Ensure that each sentence is clear and concise. For example, you might consider rephrasing the last sentence for smoother readability.

Third, a considerable degree of conceptual confusion is caused in the Introduction, when the authors use “health literacy” instead of “health education” without providing the necessary clarifications, differences, and similarities between the two concepts. Consider adding a concise thesis statement towards the end of the introduction. This statement should clearly outline the purpose of your review and what readers can expect to gain from it. Explicitly connect the importance of literacy, especially health literacy, to the concept of a "successful transition from custody to the community." Emphasize how literacy, in this context, is crucial for informed decision-making and access to health services. Ensure that the citation information, such as the academic editor, received date, revised date, accepted date, and published date, is appropriately filled in. This information is crucial for the formal presentation of the article. When presenting statistics (e.g., individuals being 12 times more likely to die within the first two weeks of release), consider providing the sources of these statistics for transparency and credibility.

Fourth the Materials and Methods section is particularly problematic in that it does not provide a robust justification why the narrative review was used as a method instead of other options. While you briefly explain the PICo strategy, you might consider providing a bit more detail, especially for readers who may not be familiar with this framework. This could involve a brief expansion on how each element (patient, intervention, context) was defined and applied in your review. Consider briefly summarizing the key aspects of the search strategy, such as the combination of descriptors and derivations used in the search. This can provide readers with a quick overview of the scope of your search. You've mentioned that two independent reviewers performed the narrative review. You might consider adding a sentence on how discrepancies or disagreements between reviewers were resolved, such as through discussion or consultation with a third reviewer.

The discussion section is well structured and provides a thorough exploration of various health education interventions for individuals transitioning from prison, covering a range of topics.

Comments on the Quality of English Language

As previously commented.

Author Response

Dear Reviewer 2,

We would like to express our sincere thanks for the time and effort you have dedicated to reviewing our article for publication in Healthcare Journal. Based on your comments, we have made the following changes to the manuscript:

-We corrected the indicated punctuation errors (please see lines 22-24);

- The beginning of the summary has been reworded, and sentences have been added to provide a broader context about the challenges faced by individuals in the transition from prison to the community and the importance of health education in this context. The WHO concept was excluded to avoid common sense at the beginning of the manuscript (please see lines 11-14);

- The main characteristics and advantages of the narrative review for this specific study were explained (please see lines 73-76)

- The main themes that emerged from the studies were briefly highlighted in the summary: HIV and other sexually transmitted infections, alcohol, opioids and other substances, tuberculosis and women´s health (please see lines 24-26).

- At the end of the summary, we add the argument of the potential impact of the review on informing future health education initiatives and policies for individuals transitioning from prison (please see lines 28-30).

- The importance of literacy, especially health literacy, was explicitly connected to the idea of ​​“successful transition from custody to community.” Emphasize how literacy, in this context, is crucial for informed decision-making and access to health services (please see lines 70-75).

- The “Material and methods” section was reviewed in detail, according to the notes of reviewer 2: a justification for using narrative review as a method was included (please see lines 73-76), and more information about the PICO strategy was provided, too (please see lines 81-86). The researchers' disagreements over whether an article should be included in the review were resolved by mutual agreement (please see lines 91-92)

Reviewer 3 Report (Previous Reviewer 2)

Comments and Suggestions for Authors

This is a reasonable review, but needs the following-

1. More details on methods.

2. Dates for inclusion and exlcusion of articles.

3, Major implications for research and practice.

4. Recommendations for practice.

5. Expansion of limitations of this work.

Comments on the Quality of English Language

Minor editing needed for this paper.

Author Response

Dear Reviewer 3,

We would like to express our sincere thanks for the time and effort you have dedicated to reviewing our article for publication in Healthcare Journal. We have made the following changes to the manuscript, based on the five major recommendations: More details about the method were added (please see lines 82-85), such as the justification for choosing the narrative literature review, the components of the PICO strategy (please see lines 88-93), and information about researchers' disagreements about whether an article should be included in the review were resolved by mutual agreement (please see lines 107-108). A description of the time required for the process of inclusion and exclusion of articles from the review was included, which took place between February and August 2023 (please see lines 119-120). The limitations of this work have been expanded (please see line 295). The main implications for research and practice (please see lines 28-30 and 306-308).

Reviewer 4 Report (Previous Reviewer 1)

Comments and Suggestions for Authors

Dear editor

Thank you for sending the article for a second revision. I was pleased to see that most of the revisions have been taken into account, which in my opinion significantly improves the article in conceptual terms. 

However, I still have a few concerns about the size of the table detailing all the articles. What I can suggest is that you make a table in which you identify the reference of the article, but present the results in summary form. The way you present them needs to be more summarised; the reader doesn't need so much detail. Having said that, I would also like to see greater consistency in the conclusion, especially where the authors respond to the study's objective of identifying and synthesising, and critically evaluating peer-reviewed evidence concerning health education initiatives developed during or after incarceration that people released from prison.

So what comes out of this evaluation and analysis? It's not enough to say that they're important. You have to answer: what real impact do these programmes have on promoting the health of this population?

I wish you success

Author Response

Dear Reviewer 4,

We would like to express our sincere thanks for the time and effort you have dedicated to reviewing our article for publication in Healthcare Journal. Based on your valuable recommendations, we adapted the table to present the results more concisely: the "Description of study" column was deleted, and the details in the last column (main results) were simplified, as can be seen from line 173.

Please accept our sincere thanks for the important feedback you provided regarding the shortcomings highlighted in the conclusion of the work. The review presented the real impacts of programs that promote the health of people with a history of incarceration (please see lines 305-313).

Round 2

Reviewer 2 Report (New Reviewer)

Comments and Suggestions for Authors

The authors have taken into account all of my comments and suggestions so that an amended version of their article has been now submitted to which I have not detected any further issues.

The topic of their work is important and deserves our attention.

Author Response

Dear Reviewer,

We would like to express our sincere thanks for the time and effort you have dedicated to reviewing our article for publication in Healthcare Journal. Based on the academic editor, we would like to inform that the topic “Implications for practice and research” has been moved to the Conclusions, and the last paragraph has been extended (please see lines 313- 321).

Thanks again for reviewing our submission.

Sincerely,

The authors.

This manuscript is a resubmission of an earlier submission. The following is a list of the peer review reports and author responses from that submission.

Round 1

Reviewer 1 Report

Comments and Suggestions for Authors

Dear Editor

The article, entitled Health Education Interventions for People Who Have Experienced Prison: An Integrative Review, intends to identify, synthesise and critically evaluate peer-reviewed evidence on the subject developed during or after incarceration for people who have experienced prison.

In the abstract: the authors first refer to health illiteracy and only then do they mention that the review was carried out on health promotion with this target population. The results consider that there are gaps, arguing that this type of review can be useful for promoting initiatives for people who have experienced imprisonment and has the potential to inspire new interventions in countries around the world.

It would be appropriate to start the abstract with the notion of health promotion and to emphasise the need to analyse studies that highlight health promotion in people who have experienced prison. Why is it important to promote the health of these people who are incarcerated or who have already had this experience? It is important to mention in the abstract which protocol was used to carry out the review (Scopus or another database). In terms of results, the abstract fails to justify the relevance of this literature review and what health promotion means in this context.

Introduction: The first paragraph begins with a definition of health literacy, mentioning that it has gained prominence in public health. Perhaps it would be relevant to first define the concept of health promotion and its determinants, and then the dimensions associated with it: literacy, public health and this target audience. Example of health promotion programmes in prisons, example of needle exchange, condom use…etc

I'm mentioning this because the title of the article is silent on the concept of health literacy. So there seems to be a disconnect between the title and the subject of the article.

In the introduction, is it also necessary to explain why you are interested in the topic? Is it part of a research project?.

Methodology: is it important to explain the inclusion criteria more fully, and note only explaint the selected articles were primary studies, using qualitative and/or quantitative..., and wat about the main key words for making this research?

It is important that the search protocol is explained, considering that you searched multiple databases, it would be important to explain in no uncertain terms which words you chose to search, whether you searched by title or by subject, and whether you searched journal articles, chapters, books, in English or other languages, etc. .... This must be clear and unambiguous.

Some of these questions are explained in Figure 1, but they lack more detail.

Results: the results are presented in a table followed by a discussion. The results should emphasise examples of health promotion programmes in prisons....

The discussion reveals aspects that justify the study and that should be summarised in the problematisation of the article, in the introduction. The discussion should highlight the importance of health promotion programmes in prisons, including those that focus on public health promotion, health literacy and other relevant areas that have been mentioned.

I think that the discussion lacks theoretical foundations, which are missing from the introduction, as I have already mentioned. The discussion is descriptive and needs to be analytical in order to respond unequivocally to the aim of the article.

I suggest clarifying the concepts of health promotion and justifying the personal and theoretical relevance of the article. It is also important to clarify the literature review scoping protocol and highlight some examples of health promotion programmes for this target population. Having said that, the results should be more analytical and less descriptive, and the concussion should unequivocally answer the starting question for the article.

Reviewer 2 Report

Comments and Suggestions for Authors

1. The paper is ridden with errors of grammar, spelling, and syntax.

2. it is a very superficial attempt to understand a broad issue.

3. Authors need to decide if they want to use PRISMA guidelines and what type of review would they call this (scoping? narrative? systematic?)

4. Health education is a very broad term and not all intervention studies relevant to this review have used that term. So, you have missed dozens of interventions that should have been included.

5. The major implications for practice and research are no where to be seen in the paper.

6. The synthesis of the literature is only a verbatim repeat of results of other studies, but no critical evaluation has been done.

Comments on the Quality of English Language

The paper is ridden with errors of grammar, spelling, and syntax.

Reviewer 3 Report

Comments and Suggestions for Authors

Dear Authors

Overall the manuscript entitled "Health Education Interventions for People Who Have Experienced Prison: An Integrative Review" is well written. 

I have some suggestions:

- Your aim or your research question is "What are the existing health education initiatives developed for adults (adults over the age of 18) recently released from prison or about to be released?” In PICo, context (Co) refers to life after the prison experience.

If you include persons "about to be released", how would you assess the life after the prison experience? Even your search terms does not reflect persons about to be released. Clarify.

- In table 1, add these details: Number of participants in each study, mean age, number of male and female, duration of study. I feel these details will give more impact and will give more information about each studies at a glance.

- What is the limitation of your review? If you find any of the included studies having less sample size, you can mention that in the limitation.